# Utilizing Cognitive Training to Improve Working Memory, Attention, and Impulsivity in School-Aged Children with ADHD and SLD

**DOI:** 10.3390/brainsci12020141

**Published:** 2022-01-21

**Authors:** Grahamm M. Wiest, Kevin P. Rosales, Lisa Looney, Eugene H. Wong, Dudley J. Wiest

**Affiliations:** 1Department of Psychology, Alliant International University, Alhambra, CA 91803, USA; gwiest12@gmail.com; 2Division of Behavioral and Organizational Sciences, Claremont Graduate University, Claremont, CA 91711, USA; kevin.rosales@cgu.edu; 3Department of Child Development, California State University, San Bernardino, CA 92407, USA; lisa.looney@csusb.edu (L.L.); office.wiest@gmail.com (D.J.W.)

**Keywords:** cognitive training, ADHD, network analysis, school setting

## Abstract

Students’ use of working memory (WM) is a key to academic success, as many subject areas and various tasks school-aged children encounter require the ability to attend to, work with, and recall information. Children with poor WM ability typically struggle with academic work compared to similar-aged peers without WM deficits. Further, WM has been shown to be significantly correlated with inattention and disorganization in those with ADHD, and WM deficits have also been identified as a potential underpinning of specific learning disorder (SLD). As an intervention technique, the use of computerized cognitive training has demonstrated improved attention and working memory skills in children with WM deficits, and children that have completed cognitive training protocols have demonstrated performance improvements in reading and math. The current study aimed to examine the effectiveness of cognitive training (conducted in a clinical setting) for students diagnosed with ADHD and SLD. Using paired-samples t-tests and a psychometric network modeling technique, results from data obtained from a sample of 43 school-aged children showed (1) that attention and working memory improved following cognitive training and (2) that cognitive training might be related to cognitive structural changes found pre- to post-training among the variables being measured. Implications for clinical practice and school-based interventions are discussed.

## 1. Introduction

Working memory (WM) refers to the active, top-down manipulation of information held in short-term memory [1] and includes distinct components including the central executive (responsible for active manipulation of stored information), the phonological loop (responsible for short-term storage and rehearsal of verbal/auditory information), and the visuospatial sketchpad (which is responsible for short-term storage and rehearsal of visual/spatial information) [1]. Throughout development, WM manifests in mental processing in a variety of ways [1,2]. For school-aged children, WM becomes a key to academic success, as many subject areas (e.g., reading, writing, spelling, and math) and various tasks school-aged children encounter (e.g., remembering and meeting deadlines; following multi-step instructions) require the ability to attend to, work with, and recall information [2,3,4,5].

### 1.1. Working Memory and Academic Skills

The role of WM in the classroom has received much attention in the literature, demonstrating the importance of WM processes for critical academic skills such as math and reading [6,7]. For instance, it has been well documented that the components of WM are involved in children’s performance on addition and subtraction tasks, as the use of mental manipulation or images of numerical digits requires the use of visuospatial resources [8,9], while counting skills are related to verbal resources through the phonological loop [10]. Further, word decoding skills and reading comprehension are also related to WM, as reading tasks require holding information (e.g., letter sounds, words, and word meanings) in short-term memory while continuing to extract additional information from upcoming text [11]. Successful reading ability (and its established association with WM skills) extends into success in other subject-area content, as children in mid-late elementary school and beyond begin to rely on reading as a primary source of information gathering for content in all areas of study [12]. Given the reliance on WM resources for successful completion of critical academic tasks, it follows that children who demonstrate good WM skills tend to also show strong levels of academic success, a conclusion demonstrated in the literature [3,13,14,15].

As children move through the early schooling years, their WM performance increases [16]. Children transition from a heavy reliance on visuospatial resources to an increased ability to convert information to a phonological code and utilize verbal resources, likely as a result of their increased language and reading proficiency [14,16,17]. In addition, increased attentional capacity, higher rates of processing speed, and increased content knowledge in long-term memory also contribute to higher levels of WM performance [16]. These strengthened WM skills assist children in academic tasks as they progress through school, since school tasks, responsibilities, and content extend in size and difficulty throughout the school years.

The developmental nature of WM resources is of particular note, as studies have shown that WM skills measured before the start of formal schooling predict academic performance in subsequent academic years [3,14,18] and these skills serve as an even more powerful predictor of later academic success than IQ in the early school years [3]. In short, WM skills are critical to current and future academic success. Therefore, findings such as those discussed are significant as they underscore the importance of early identification of difficulties and implementation of intervention, so as to not deter learning of basic, foundational skills in the early grades [14,19] that will ultimately impact academic performance in later grades.

### 1.2. Working Memory Deficits

While WM relates to academic performance, variation occurs in WM functioning across individuals [20]. Even when children are provided adequate and effective academic foundations, deficits in WM can impede academic progress [2,5,21], creating frustrations for students, parents, and educators [22,23,24]. Children with poor WM ability typically struggle with academic work compared to similar-aged peers without WM deficits [25]. Evidence has linked WM capacity to the ability to learn novel ideas in both reading [7] and mathematics [6,26,27]); however, impairments in verbal or visuospatial WM have been associated with poorer writing and reading skills [26,28,29], spelling difficulties [29], and math performance [6,26], in addition to struggles in classroom activities such as following instructions or skipping steps in complex tasks [2] and higher identification of problem behaviors by teachers [21,30].

Deficits in WM have been documented in approximately 10% of children in mainstream classrooms [30]; a percentage significant enough to warrant efforts to bolster WM for school-aged children. As has already been established, WM supports tasks that children routinely encounter in the school environment, and given the associations found between WM deficits and poor academic achievement, these deficits put school-aged children at risk for learning disabilities [30], suggesting that an emphasis toward the improvement of WM could help to ameliorate learning issues for some children.

### 1.3. Working Memory and Clinical Populations

*Attention Deficit Hyperactivity Disorder.* While WM deficits are a matter of importance, in general, for school-aged children, concerns are noticeably greater for children diagnosed with attention deficit hyperactivity disorder (ADHD), in particular. WM has been shown to be significantly correlated with inattention and disorganization in those with ADHD [31,32]. Attention is a crucial component of WM [33,34], since one must first attend to information to store, manipulate, and retrieve it. Therefore, it follows that children with a diagnosis of ADHD often experience WM deficits [19,35,36,37,38]. Structured learning demands often overload WM in these populations, resulting in a loss of critical information needed for task completion [2].

ADHD is characterized by difficulties with hyperactivity, impulsivity, and sustained attention [39] and affects approximately 3% to 7% of the childhood population [40]. In many ways, the symptoms of ADHD hamper academic achievement [41], and core symptoms such as inattention and behavioral dysregulation are the main culprits in creating disruption in the classroom [41]. Students with ADHD (as well as those with WM problems without an ADHD diagnosis) present with symptoms of inattentiveness and are often characterized by educators as being off task or as poor listeners, rather than depicted as students with memory problems [13]. Gathercole [13] argues that inattentiveness is a symptom of WM deficits experienced by these children, and that the overload of WM processes that results in the loss of information needed to complete a task shifts attention away, leading WM deficits to be misinterpreted as lack of attention. Experimental studies have suggested a potential causal relationship between WM difficulties and inattentive and hyperactive behavior brought about by ADHD [32]. Further, studies have documented a decrease in ADHD symptoms in children who demonstrated age-related improvements in WM [36].

*Specific Learning Disorder.* WM deficits have also been identified as a potential underpinning of specific learning disorder (SLD), and children with this diagnosis present with identified academic struggles in reading, writing, and/or math [42]. Similar to children with ADHD, problems in WM processing have been documented in children experiencing a SLD. Specifically, deficits in phonological processing [4], inhibited central executive functioning [43], and difficulties with sustained attention have been associated with SLD diagnoses [44].

Enhancing WM skills in ADHD and SLD populations has significant potential implications. Given the established importance between WM and academic success, finding ways to intervene and improve WM in those with deficits (such as within clinical populations with ADHD and/or SLD) could help to improve academic performance.

### 1.4. Cognitive Training: An Approach for Enhancing Working Memory

Relationships have been established between WM and academic performance, as well as between WM and the inattention, poor inhibition, and mental organization of children with ADHD [40] and SLD [42], resulting in school-aged children with ADHD and/or SLD exhibiting difficulties with academic performance [3,43,45]. In order to improve academic performance through attention, inhibition, and mental organization, treatment protocols for children (particularly those with ADHD) have consisted of behavioral interventions, psychostimulant medications, or a combination of both [40,46], with varying degrees of efficacy. However, another intervention approach—computerized cognitive training—has shown promising results. Specifically, studies have demonstrated that the use of computerized training protocols has been associated with improved attention and visual and auditory WM [47,48,49]. In the last couple of decades, as evidence about the effectiveness of such programs has made its way into the literature, increased attention has been given to how these programs can be effectively used for school-aged children within the school or classroom setting [50].

Computerized cognitive training involves the use of brain games, targeting different cognitive skills, including attention, concentration, verbal and visual WM, processing speed, and inhibition [47]. Using a model that is adaptive in nature (i.e., the activity increases or decreases in difficulty, depending upon a student’s performance), cognitive training programs have demonstrated performance gains in various cognitive and WM tasks after 20 h of intervention [47,50,51], with maintained improvements observed over a six-month period [52,53]. Specifically, the use of n-back tasks (those that require individuals to examine whether a stimulus is the same as one presented in a previous trial) that address aspects of WM have been shown to decrease WM deficits over time [49].

### 1.5. The Use of Cognitive Training in Schools

Increased awareness of the educational difficulties experienced by those with deficits in WM has led to higher demands for educational interventions. Computerized cognitive training offers one potential intervention approach. Studies examining computerized cognitive WM training have revealed promising results, demonstrating greater performance improvements in such academic skills as reading [54,55] and math [54,56]. Variability exists in effectiveness of transfer effects (from WM training to academic performance) depending on such factors as duration of training [57]), baseline performance [55], sleeper effects [58], supervision during training [57], the addition of game elements to training tasks [55], motivation [59], and the types of academic skills measured [54,60].

While variability across studies exists due to the factors mentioned, another potential influence is the setting in which training takes place. One limitation to existing cognitive training research is the lack of examination of children in school or clinical settings. In other words, a considerable amount of work in this area has examined training effects that occur outside of a more controlled setting. One common methodology across cognitive training studies is to conduct pre- and post-test analyses on WM tasks, while the actual intervention occurs in the home environment (e.g., training sessions took place for specified time intervals in the home, outside of clinical supervision). Methodologies that examine cognitive training protocols within school or clinical settings (with the continuous presence of trained examiners) are needed, as this might provide additional information related to the effectiveness of this approach.

### 1.6. The Current Study

The current study’s aim was to examine the effectiveness of cognitive training for students diagnosed with ADHD and a co-occurring learning disorder. Of particular note, this project was conducted in a clinical setting; thus, students were pre-tested for baseline cognitive abilities, received training, and post-tested for the same cognitive abilities in a more controlled environment. All hypotheses are stated directionally based upon previous work, e.g., see [50]; 

**Hypothesis** **1** **(H1):**
*it is expected that inhibition capabilities will improve after cognitive training;*


**Hypothesis** **2** **(H2):**
*it is expected that attention capabilities will improve after cognitive training;*


**Hypothesis** **3** **(H3):**
*it is expected that overall working memory capabilities (verbal and visual working memory) will improve following cognitive training;*


**Hypothesis** **4** **(H4):**
*it is expected that verbal working memory will improve after cognitive training;*


**Hypothesis** **5** **(H5):**
*it is expected that visuospatial working memory will improve after cognitive training.*


Further, this study examined the underlying structure of cognitive abilities using a more novel psychometric technique: network analysis. This analysis uses observed variables and the partial correlations between those observations. Importantly, network analysis does so without assuming latent common causes (which is the case with traditional latent variable models). Instead, network analysis conceptualizes psychological constructs as connected networks. In these models, the observed variables are referred to as nodes and the connections between them are referred to as edges. This approach lends itself well to cognitive abilities research given that these theoretical models regard cognitive processes as dynamic and interactive [61]. Thus, this aspect of the project provides important insights into the interconnectedness of cognitive abilities which has direct implications for clinical practice and school-based interventions for students identified with learning needs.

## 2. Method

### 2.1. Research Design and Participants

This project examined the effectiveness of computerized cognitive training for enhancing working memory and attention in children and adolescents (between 6 and 17 years of age; *M* = 11.7, *SD* = 3.87) with ADHD and co-occurring SLD. Twenty-six were diagnosed with a reading SLD, 6 with a writing SLD, 4 with a math SLD, and the remaining with no SLD. This study utilized an archival data review from a local private practice clinic in southern California where the cognitive training occurred. Taken together, this project employed a quasi-experimental, within-subjects design. Pre- and post-test scores on WM, attention, and inhibition were used to evaluate the effectiveness of cognitive training. The sample (n = 43) consists of 22 males and 21 females. Additional information such as ethnicity, family income, and academic information was not initially gathered and therefore not reported here. However, all participants had received a psychoeducational evaluation because of continuing educational concerns; and each participant received a recommendation to enroll in the cognitive training as a result of their diagnoses.

### 2.2. Measures

***Integrated Visual and Auditory Continuous Perform Test, Second Edition (IVA-2)*****.** The IVA-2 is decision support software that helps clinicians test and evaluate both visual and auditory attention and response control functioning. Validity research using the IVA CPT with children ages 7 to 12 had a sensitivity rate of 92% in accurately identifying individuals with ADHD. The IVA CPT also correctly identified the 90% of non-ADHD children (i.e., false positives = 10%) [62]. Another validity study for a typical mixed age clinical population (ages 6 to 55) found that as part of a clinician’s comprehensive psychological evaluation, the combination of the ADHD rating scale data with the IVA CPT matched the clinical diagnosis 90% of the time. In addition, this study did not misclassify 89% of individuals who did not have ADHD (i.e., 11% false positives) [63].

***Wide Range Assessment of Learning and Memory, Second Edition (WRAML-2).*** The WRAML-2 is a norm-based measure of memory functioning and learning in individuals ranging from 5 to 90 years of age. In this study, the finger windows and number-letter subtests were used to examine the construct of attention, while the symbolic working memory and verbal working memory subtests measured the construct of working memory [64]. Alpha coefficients for the WRAML-2 are strong for the subtests used in this study (alpha of 0.81 to 0.84).

***Wechsler Intelligence Scale for Children, Fifth Edition (WISC-V)*****.** The WISC-V is a norm-referenced measure of intelligence for children ages 6 to 16 years. The digit span and picture span subtests were administered to assess working memory. Both subtests have strong psychometric properties; alpha coefficients of 0.93 and 0.82 were reported for digit span and picture span, respectively [65]. While the reliability and validity exchanges for the WRAML-2 have adequate psychometric properties, the WISC-V working memory subtests were also utilized due to its stronger norms with a focus on children and adolescents [64].

In summary, this study utilized several outcome measurements examining overall working memory, verbal working memory, visuospatial working memory, attention, and inhibition.

Overall Working Memory
Working Memory Index from the WISC-V which is comprised of Digit Span and Picture Span subtests
Verbal Working Memory
Verbal Working Memory subtest from the WRAML-2
Visuospatial Working Memory
Symbolic Working Memory subtest from the WRAML-2
Attention
Attention Index from the IVA-2
Inhibition
Response Inhibition Index from the IVA-2


### 2.3. Procedures

Two criteria were set for participation in this study: (a) a diagnosis of ADHD and SLD and (b) a resulting clinical recommendation for a cognitive training program. Parents who chose to have their children participate in the training program provided consent for their children. Baseline data, or pre-treatment data were gathered shortly before the commencement of the training intervention. 

**Intervention.** Each child received cognitive training via the Captain’s Log program. Captain’s Log is a suite of computer-based games (developed by BrainTrain) that target cognitive abilities. For example, one activity, Space Race, requires participants to “shoot” a laser at space obstacles as they travel through space. Alternatively, “Fire Dragon” requires participants to protect a castle wall from a dragon and an opposing knight. A recent meta-analysis conducted by Rossignoli-Palomeque and colleagues [66] noted the valid and effective nature of Captain’s Log as a computer-based cognitive training program for remediating WM deficits in children.

The intervention program utilized in this project consisted of 20 sessions which were completed in approximately 4–8 weeks, with each session lasting 60 min. Sessions took place in a clinic setting and were supervised by a psychometrician or staff member to increase effort and motivation. Task frustration tolerance was addressed through encouragement, active praise, positive reinforcement, short breaks, rewards, and/or a token economy. One week after the 20 sessions were completed, the participants were given a post-test examination and feedback session with the parents. Training-related improvements were monitored with the same measures as the pre-test analysis which included verbal working memory, symbolic working memory from the WRAML-2; digit span and picture span subtests from the WISC-V; and the IVA-2 continuous performance task.

### 2.4. Data Analysis

Each participant was given a face sheet which tracked their pre-treatment scores, session date, number of sessions, and post-training scores. This archival information served as the data for this project.

Paired-sample t-tests were used to evaluate each of this project’s hypotheses. These hypotheses were driven by the current literature pertaining to cognitive training for children with academic impairments. More specifically, each hypothesis utilized in this study was directed by research conducted with children with ADHD and SLD. Within this context, children with ADHD and SLD have shown improvement in cognitive abilities (e.g., WM) following cognitive training; thus, all hypotheses were directed toward positive changes after training. 

Cohen’s d, was used to determine the magnitude of the effect of treatment. Generally speaking, an effect size of 0.2 is considered small, 0.5 represents a medium effect, and 0.8 a large effect size [67]. This study included 43 participants which is sufficient for statistical power.

*Network analysis.* Psychometric network models were conducted on the post-test correlation matrix for the current study. Analyses were conducted using the qgraph and open MX packages in R. Results were visualized using qgraph. The network models were produced using the graphical least absolute shrinkage and selector operator (gLASSO) regularization technique [68]. Two parameters were manually set: hyperparameter gamma (set to 0.50) and the tuning parameter lambda (set to 0.01). Setting the hyperparameter conservatively as in this study allows for the regularization technique to prefer simpler models with fewer edges. The lambda parameter settings set here allowed for only the detection of true edges and not any spurious edges.

## 3. Results

Data for forty-three school-aged children (22 female) were contained in the archival record. However, the number of children who fully completed each assessment varied. There were 43 students who completed the attention and inhibition assessment. Thirty-one students completed the overall working memory assessment. Twenty-eight total students completed the visuospatial task, and 27 students completed the verbal working memory task. There were several reasons for this variability. On the IVA-2, if a student scored below a certain threshold, their score became invalid, and a standard score was not given. Additionally, differences in the N on specific measures were attributed to either selection survival or attrition.

All full-scale scores were represented by standard scores (M = 100; SD = 15; effective range = 69–131) while individual subtests scores were represented by scaled scores (M = 10, SD = 3; range = 0–19). Table 1 shows the mean, number of students, and the standard deviation of scores.

Figure 1 shows the average *scaled scores* pre- and post-training for the verbal and visuospatial working memory tasks, while Figure 2 shows the mean in *standard scores* pre- and post-training for inhibition, attention, and overall working memory tasks. The reason for differentiating between scaled scores and standard scores was because overall working memory, attention, and inhibition indices were recorded using standard scores, while the WRAML-2 subtests such as verbal working memory and symbolic working memory utilized scaled scores.

Paired-samples t-tests were conducted to evaluate the impact of Braintrain on students’ inhibition, attention, overall working memory, verbal working memory, and visuospatial working memory. Overall, paired comparisons supported the hypothesis that cognitive training improves working memory, attention, and inhibition.

The inhibition index (Hypothesis 1) was significantly higher from pre-assessment (M = 71.6, SD = 38.64) to post-assessment (M = 93.25, SD = 26.76), t(39) = 4.35, p = < 0.00 with a large effect size (d = 0.65).

The attention performance (Hypothesis 2) was significantly higher from pre-assessment (M = 68.39, SD = 40.69) to post-assessment (M = 86.37, SD = 26.42), t(40) = 3.06, *p* = 0.004 with a medium effect size (d = 0.52).

Overall working memory (Hypothesis 3) was significantly higher from pre-assessment (M = 86.55, SD = 8.17) to post-assessment (M = 101.10, SD = 9.1), t(30) = 8.40, *p* < 0.00. with a large effect size (d = 1.68).

The verbal working memory scores (Hypothesis 4) were significantly higher from pre-test (M = 10.11, SD = 1.80) to post-test (M = 11.11, SD = 2.28), t (26) = 2.70, *p* = 0.012 with a small effect size (d = 0.48).

Lastly, the visuospatial working memory (Hypothesis 5) scores were significantly higher from pre-test (M = 9.35, SD = 1.89) to post-test (M = 10.03, SD = 1.48), t (27) = 2.14, *p* <0.05 with a small effect size (d = 0.41).

### Network Model Results

Although the original intent was to produce two network models (one for pre-training data and one for post-training results), it was not possible to establish a model for the pre-training data because of the lack of intercorrelations among the pre-training variables included in this project. See Table 2 and Table 3 for correlations among pre-test and post-test variables. This suggests that (prior to training), there was a lack of connectedness among the cognitive abilities measured although one would expect notable correlations among working memory, attention, and inhibition. Following cognitive training, the correlations among the measured constructs were sufficiently strong to conduct a network analysis. The change in connectedness of cognitive abilities pre-training to post-training is particularly worthy of note, as it suggests that the cognitive training might be related to structural changes among the cognitive variables being measured (see Section 4 for further analysis of this finding).

Overall, the network model analyses showed an underlying structure of cognitive abilities, where the working memory tasks, attentional tasks, and inhibition tasks each form their own respective clusters. This shows appropriate construct validity as well as convergent validity. As can be seen in Figure 3, the underlying structure of cognitive abilities in school-aged children with ADHD and SLDs is seemingly well defined. The WM, inhibition, and attention constructs are clearly separate (but related), and thus are compatible with current models of cognition. However, it is important to note that model fit indices for the current network model do not indicate appropriate fit. See Table 4 below. Nonetheless, clear clusters can be observed and provide important information regarding the psychometric structure of these abilities. 

## 4. Discussion

This project had two primary purposes. First, the efficacy of cognitive training (implemented within a clinical setting) for students who were diagnosed with ADHD and a learning disorder was examined. Second, this project examined the underlying structure among the included cognitive abilities. Overall, the findings support the effectiveness of cognitive training in improving specific cognitive abilities within a clinical sample. Further, the current project demonstrates structural changes following 20 hours of computer-based intervention.

### 4.1. Pre- to Post-Training Improvement in Cognitive Abilities

Working memory is the ability to temporarily store, manipulate, and retrieve information while completing cognitive tasks [1]. Working memory deficits are commonly seen in children with attention-deficit/hyperactivity disorder (ADHD) and specific learning disorders (SLD) [19,32,43]. Working memory is linked to reading and mathematics achievement, making it an essential component for the academic success of school-aged children [6,7]. Deficits in working memory may lead children to fall behind their peers as well as miss important academic benchmarks in their school experience [3]. Being attentive is an important precursor to effective WM [33]. It is, of course, a core symptom of the inattentive type of ADHD. 

Within groups paired-samples t-tests demonstrated that attention and working memory were improved following a 20 hour individually administered cognitive training program. Specifically, participants’ scores in attention, inhibition, overall working memory, verbal working memory, and visuospatial working memory were significantly higher after the intervention. Associated effect sizes ranged from small to large. These findings are consistent with a growing body of research that documents the effective use of cognitive training in ameliorating deficits in certain cognitive abilities (e.g., see [31,47,48,49,50]. Further, a number of meta-analyses have shown that cognitive training is effective in enhancing cognitive abilities, especially among clinical populations, e.g., see [69].

It is important to note that the program (Captain’s Log) utilized in this project is adaptive in nature. That is, the level of challenge that a participant experiences at any point is calibrated to their performance. Thus, the current task should not be too easy nor too difficult. This characteristic of the training is especially important in a clinical setting (and with diagnosed individuals) as there is generally a need to “adjust” in real-time when providing a therapeutic experience. In fact, it may be the targeted nature of the cognitive training that is, in part, responsible for the effectiveness of the intervention itself. Future work can further address this possibility by having some participants receive an adaptive version of training while others do not.

In general, a methodological weakness associated with previous cognitive training research is that it has been conducted in a variety of settings (e.g., in the participant’s home), as opposed to formal settings such as schools and clinical practices. Thus, this project offers important insights into the potential value of cognitive training as it focused on a clinically diagnosed sample that received the intervention within a clinical practice. In this study, participants worked one-on-one with a technician while completing the training activities during after-school hours. In having a technician present at all times, it was possible (a) to verify that the participant understood the “game” they were playing, (b) to offer support to the participant (as needed), and (c) to address any issues with attentiveness/engagement during each session.

Ultimately, the pre- to post-training differences that were documented in this clinical sample provide further evidence for near-transfer effects (i.e., train a specific skill/ability, see a change in that ability) in the cognitive training literature. This is clearly an important finding within the context of a clinical sample as we expect to observe a “therapeutic effect” in the clinic. However, future work (both in clinical settings and school environments) should also address far transfer effects (i.e., skills not directly trained demonstrate a change as a result of an intervention); this can be referred to as generalization of training. In both clinical practice and schools, such an effect has important implications for typical outcomes of interest (e.g., academic performance).

### 4.2. Underlying Structure of Cognitive Abilities Prior to and following Training

In regard to the second aspect of the current study, this project used a novel exploratory technique, network analysis, to examine the underlying cognitive structure in school-aged children with ADHD and SLDs. As reported, the current set of results shows a clear structure of cognitive abilities (for the post-training data) in which the constructs of WM, attention, and inhibition are distinct but related cognitive constructs. In line with previous research [70], the current findings show a cohesive structure between WM, attention, and inhibition. This corroborates decades of research on the executive attention theory of WM [71] that postulates that attention is a central component to variation in WM abilities. However, the inability to establish a network model for the pre-training data was unexpected but could be an especially important finding. Specifically, as noted above, there is considerable work that establishes interconnections among WM, attention, and inhibition; the lack of such correlations in the pre-training data with this learning differences sample suggests that cognitive abilities that should “operate together” are not, in fact, doing so. This may contribute to (or explain) some of the real-time challenges that students with ADHD and/or SLD experience in the school setting. Further, because a clean network model could be established for the post-training data (because there were significant correlations among the construct), there is some evidence for true effects associated with the cognitive training program. It will be very important for future research to examine a number of questions. First, do other samples of students with learning differences also show a lack of intercorrelations among WM, attention, and inhibition? Do typically developing samples of students demonstrate these correlations? Do intercorrelations emerge among WM, attention, and inhibition following training that facilitate a specification of a network model that is consistent with previous research documenting the interrelated nature of these constructs? Addressing these questions has important implications for cognitive research as well as clinically oriented and school-based work with specific student populations.

It is also worth noting the tight cluster formed by the attention measures. Importantly, attention seems to form one cluster independent of the domain-specificity of the measures. In other words, the verbal and visual measures of attention and sustained attention do not form their own domain-specific clusters. This is important for both theoretical and practical purposes. Specifically, this finding shows that for selection of measures, the distinction between auditory and visual is not critical. One can administer either of the measures and still be assessing the same underlying ability. This is important given the limited amount of time and financial resources that sometimes is a significant consideration in the clinical and school settings.

Ultimately, this is one of the first studies to implement network analysis with these kinds of data and with this specialized population. We see initial evidence here for a cohesive structure of cognitive abilities in school-aged children with ADHD and SLDs following cognitive training. Since this is among the first studies to document this type of evidence, future studies of the same nature should implement network analysis as a psychometric model to continue to explore the nature of cognitive abilities in typical and atypical populations. Addressing this issue has important implications for clinical assessment. 

### 4.3. Limitations

Taken together, the current study uses a quasi-experimental and psychometric approach to understanding the effects of cognitive training and to explore the nature of cognitive abilities in a clinical sample. We corroborate previous findings that demonstrate the effectiveness of cognitive training while also providing novel evidence of the underlying psychometric structure of cognitive abilities using network analysis. However, it is also important to highlight potential limitations in the study design. It should be noted, for instance, that the current study’s methodology did not utilize a control group. While the use of a control group would strengthen an argument for the efficacy of cognitive training on working memory and attention outcomes, the current study aimed to examine the effect of cognitive training in a clinical sample, which raises some concerns for the use of a control group. For instance, the field has noted possible ethical implications of standard no-treatment control groups among clinical populations, highlighting the denial of possible “best treatment” options for those placed in comparison groups [72]. Relatedly, clinicians have discussed the use of active comparison control groups as an alternative [72]; however, in the case of cognitive training, any alternative form of treatment could potentially still impact an individual’s working memory and attention, thereby diminishing the different form of treatment as a true comparison.

The lack of control group also raises the concern of possible practice effects as an explanation for study outcomes. Calamia and colleagues [73] highlight this concern, as increases in performance outcomes might be due to factors such as memory of testing items and perfection of testing strategies. However, research has highlighted the inconsistency of practice effects, demonstrating that such factors as test construction, age of participants, neurological status of participants, and time between tests can alter the presence of practice effects [73]. Given (1) the issues surrounding the use of control groups in clinical samples and with cognitive training (as discussed above), (2) the inconsistencies present in practice effects, and (3) the lack of control groups in much of the cognitive training literature to date, this study falls in line with those that have come before in the field. Nevertheless, future research should consider ways to mitigate these limitations in order to unequivocally highlight cognitive training as an effective intervention.

### 4.4. Conclusions

This project presents findings from a clinical application of cognitive training among students identified with an attention disorder and/or a specific learning disability. Notwithstanding the limitations discussed above, the current results add to the growing literature that supports the efficacy of cognitive training in building specific abilities (e.g., working memory and attention). For example, de Oliveira-Rosa et al. [47], Gray et al. [48], and Jones et al. [49] have each reported increased WM capabilities following training. Recently, Wiest et al. [50] among others have demonstrated the positive impact of cognitive training in the school setting. Thus, there is promising evidence for the use of computer-based programs to build cognitive abilities that predict academic performance. Such results have important implications for clinical practice and the development of interventions for students. Furthermore, the current work provides some evidence for the underlying structure among multiple cognitive skills using a novel analytical technique (i.e., network analysis). Demonstrating the interconnectedness of cognitive abilities in this manner provides important insights into how these construct are related to one another and substantiates previous cognitive research. Additionally, the network analysis has potential clinical significance in that it can inform intervention/remediation protocol development.

## Figures and Tables

**Figure 1 brainsci-12-00141-f001:**
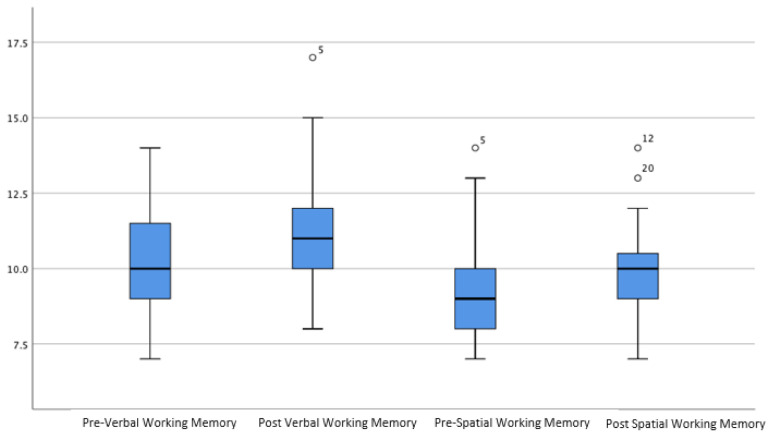
The mean in scaled score pre- and post-training for verbal and visuospatial working memory tasks.

**Figure 2 brainsci-12-00141-f002:**
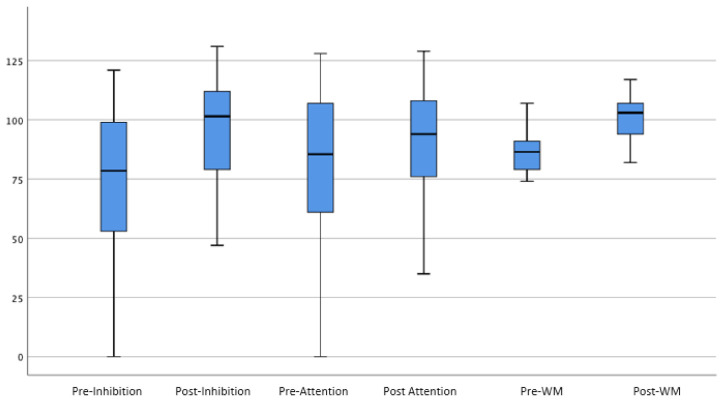
The mean in standard score pre- and post-training for the inhibition, attention, and overall working memory tasks.

**Figure 3 brainsci-12-00141-f003:**
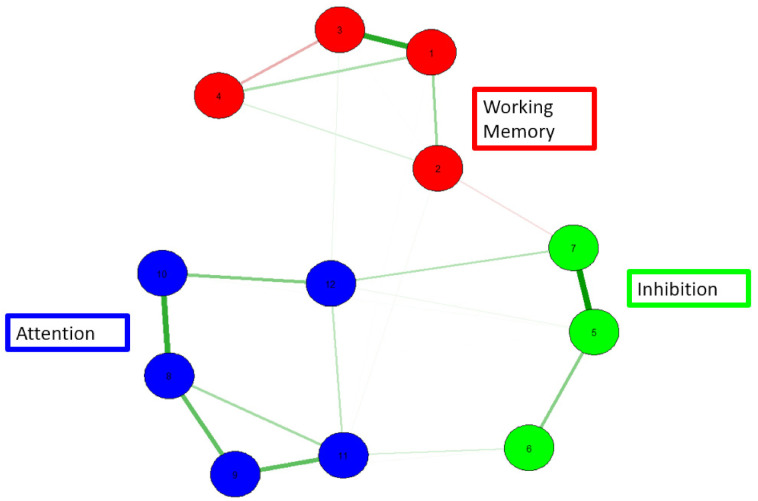
A network model of working memory, attention, and inhibition. 1 = verbal WM, 2 = symbolic WM, 3= digit span, 4 = picture span, 5 = full inhibition, 6 = auditory inhibition, 7 = visual inhibition, 8 = general attention, 9 = auditory attention, 10 = visual attention, 11 = auditory sustained attention, and 12 = visual sustained attention. Red = WM; green = inhibition; blue = attention.

**Table 1 brainsci-12-00141-t001:** Descriptive statistics for all cognitive measures.

Assessment	N	Mean	Std. Deviation
IVA-2 Response Inhibition Pre-Test	40	71.6	38.64
IVA-2 Response Inhibition Post-Test	40	93.25	26.76
IVA-2 Attention Pre-Test	41	68.39	40.70
IVA-2 Attention Post-Test	41	86.37	26.43
WISC-V Working Memory Index Pre-Test	31	86.55	8.17
WISC-V Working Memory Index Post-Test	31	101.10	9.11
WRAML-2 Verbal Working Memory Pre-Test	27	10.11	1.80
WRAML-2 Verbal Working Memory Post-Test	27	11.11	2.28
WRAML-2 Symbolic Working Memory Pre-Test	28	9.35	1.89
WRAML-2 Symbolic Working Memory Post-Test	28	10.03	1.48

**Table 2 brainsci-12-00141-t002:** Correlations among pre-test measures of cognition.

Measure	1	2	3	4	5	6	7	8	9	10	11	12
VWM (1)	-											
SWM (2)	0.46	-										
Digit Span (3)	0.22	0.28	-									
Picture Span (4)	0.17	0.10	−0.22	-								
Response Control (5)	0.09	−0.29	−0.20	−0.14	-							
Auditory Response Control (6)	0.10	−0.35	0.00	−0.02	0.75	-						
Visual Response Control (7)	0.07	−0.28	−0.15	−0.20	0.97	0.62	-					
Full Attention (8)	0.15	−0.07	−0.20	0.01	0.88	0.64	0.83	-				
Auditory Attention (9)	0.15	−0.15	0.00	−0.04	0.66	0.81	0.61	0.80	-			
Visual Attention (10)	0.10	−0.06	−0.17	−0.01	0.88	0.63	0.84	0.97	0.71	-		
Auditory Sustained Attention (11)	0.12	−0.13	−0.03	0.16	0.70	0.76	0.64	0.81	0.89	0.75	-	
Visual Sustained Attention (12)	0.25	0.05	−0.09	0.01	0.87	0.63	0.84	0.90	0.65	0.94	0.75	-

VWM = visual working memory, SWM = symbolic working memory.

**Table 3 brainsci-12-00141-t003:** Correlations among post-test measures of cognition.

Measure	1	2	3	4	5	6	7	8	9	10	11	12
VWM (1)	-											
SWM (2)	0.48	-										
Digit Span (3)	0.70	0.32	-									
Picture Span (4)	0.29	0.28	−0.21	-								
Response Control (5)	−0.10	−0.20	0.01	−0.01	-							
Auditory Response Control (6)	−0.10	−0.09	−0.08	−0.02	0.65	-						
Visual Response Control (7)	0.05	−0.25	0.14	−0.03	0.91	0.43	-					
Full Attention (8)	0.07	0.10	0.14	0.04	0.49	0.38	0.34	-				
Auditory Attention (9)	0.03	0.08	0.08	−0.01	0.38	0.45	0.21	0.93	-			
Visual Attention (10)	0.09	0.12	0.20	0.09	0.51	0.27	0.42	0.94	0.77	-		
Auditory Sustained Attention (11)	0.20	0.16	0.13	0.11	0.47	0.49	0.31	0.89	0.91	0.78	-	
Visual Sustained Attention (12)	0.11	0.00	0.26	0.06	0.61	0.36	0.64	0.72	0.58	0.79	0.71	-

**Table 4 brainsci-12-00141-t004:** Model fit indices for network model of post-test measures.

Model	χ^2^	df	CFI (TLI)	RMSEA	AIC(BIC)
Network	156.76 ***	42	0.81 (0.70)	0.25	1084 (228)

*Note.* *** *p* < 0.001; χ^2^ = model chi-square value; df = degrees of freedom; AIC = Akaike information criteria; BIC = sample size-adjusted Bayesian information criteria; CFI = comparative fit index; TFI = Tucker–Lewis fit index; RMSEA = root mean square error of approximation.

## Data Availability

Data available upon request.

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
