# Peer review of "Utilizing Cognitive Training to Improve Working Memory, Attention, and Impulsivity in School-Aged Children with ADHD and SLD"

_brainsci, 2022, doi:10.3390/brainsci12020141_

Round 1

Reviewer 1 Report

It is suggested to integrate literature on computer training for ADHD and other learning disorders as well as on computer training in schools.

In the table with the descriptive statistics of the tests, enter the minimum and maximum score for each test. Specify how many children for each age group as there is a wide range of children and adolescents involved.

Provide some examples of BrainTrain program activities

The paragraph on the limits of research is missing; an important limitation such as the absence of the control group or the wide range of subjects involved is not reported.

471 is missing parentheses

Reviewer 2 Report

The paper reports the effectiveness of an intervention applied in a clinical setting for children (age 6 to 17 years) and finds the interventions work. Although the layout is correct and well-written, the scientific soundness is missing, there is no control group, the hypothesis is not motivated, developmental changes in the small sample are not taken into account, no transfer measures are reported and no corrections for multiple testing are applied. Furthermore the power analysis is not valid given that some t-tests contained only 27 participants and not the whole sample of 43. I think the paper is of merit for clinicians working with mentioned program and for the authors to show to their funders/employers.

It's underpowered (the n=43 comes out of the blue, I have no idea which critical, hypothesis testing testing inferential statistics were planned to run and those are run on smaller samples), no hypotheses are motivated (only vaguely it is informed that participants are assessed in the clinic and not at home, but when trainings at home show improvements than they should do so in an controlled setting as is the clinic even more), and no correction for multiple testing are used. All of these cannot be corrected without further testing which means in case of this study to dig up more data the clinic seems have to have collected over the years. Finally, participants span 11 years of age and various diagnoses which are lumped together ignoring medication, socioeconomic status, depression and impairment severity, mostly because the sample it to small to run statistics that take into account these differences. Eleven years means a lot in development for the sample under consideration.

Round 2

Reviewer 2 Report

I did not realize any changes except the removal of the power analyses. The authors try to highlight the need for studies conducted in clinical samples under controlled conditions but I am not satisfied by their argument - anyhow, I think the paper will or will not be evaluated by the number of citations.